# Importance of Early Intervention in Reducing Autistic Symptoms and Speech–Language Deficits in Children with Autism Spectrum Disorder

**DOI:** 10.3390/children10010122

**Published:** 2023-01-06

**Authors:** Slavica Maksimović, Maša Marisavljević, Nina Stanojević, Milica Ćirović, Silvana Punišić, Tatjana Adamović, Jelena Đorđević, Ivan Krgović, Miško Subotić

**Affiliations:** 1Cognitive Neuroscience Department, Research and Development Institute “Life Activitites Advancement Center”, 11000 Belgrade, Serbia; 2Department of Speech, Language, and Hearing Sciences, Institute for Experimental Phonetics and Speech Pathology “Đorđe Kostić”, 11000 Belgrade, Serbia; 3Clinic for Neurology and Psychiatry for Children and Youth, 11000 Belgrade, Serbia; 4Department of Psychiatry, Faculty of Medical Sciences, University of Kragujevac, 34000 Kragujevac, Serbia; 5Institute for Children’s Diseases, Clinical Center of Montenegro, 81000 Podgorica, Montenegro

**Keywords:** autism spectrum disorder, indegrative therapy, early intervention

## Abstract

The intervention focused on starting treatment at an early age to develop the child’s full potential, which is known as early intervention. Given that autistic symptoms and language deficits occur at an early age and affect other areas of development in children with autistic spectrum disorder, we wanted to examine if early intervention is more effective in the reduction in autistic symptoms and language deficits in children aged 36–47 months old when compared to children 48–60 months old. The sample consisted of 29 children diagnosed with ASD who were admitted for integrative therapy. All participants were divided into two groups based on age: G1: 36–47 months old children, and G2: 48–60 months old children. To estimate the presence of autistic symptoms, we used the GARS-3, and for the assessment of speech–language abilities, we used the subscale Estimated Speech and Language Development (ESLD). Our results regarding the effect of the group on the difference in the scores at two time points showed that there was a statistically significant effect of the group on the reduction in autistic symptoms (*p* < 0.05) but no effect of the group on the differences in speech–language abilities between the two time points (*p* > 0.05). Our study highlights the importance of emphasizing the exact age when using the terms “early intervention” and “early development” in future studies and practice because it is necessary to determine and establish guidelines about which particular ages are crucial for starting treatment in certain developmental aspects.

## 1. Introduction

Autism spectrum disorder (ASD) is defined as a neurodevelopmental disorder characterized by social interaction, communication impairment, and behavioral disorders that recur with atypical or narrow interest [1]. Associated symptoms in ASD are decreased or increased sensory sensitivity, hyperactivity, attention and behavior problems, and emotional, sleep, and mood disturbances, which makes ASD a lifelong neurodevelopmental condition and its clinical picture very diverse [2]. Knowing that the manifestation of ASD can be very divergent, there is a need for more research in order to better understand the mechanisms of successful interventions and to identify all the variables important for the prediction of optimal outcomes [3].

Russell, Stapley [4] stated the rising number of detected ASDs since the 1990s can be partially explained by the increased recognition of the condition due to better diagnostics. According to Newschaffer, Croen [5], the prevalence of people diagnosed with ASD worldwide is estimated to be 1–2 per 1000 people. Many studies have outlined the growing number of people diagnosed with ASD [6,7]. This growing prevalence of ASD highlights the importance of early diagnosis and early intensive interventions for reducing the impact of symptoms on children’s functioning.

According to diagnostic criteria, ASD symptoms appear around 12 and 18 months of age; however, sensory and motor symptoms often occur earlier, during the first 12 months [8]. In most youngsters, the manifestation of ASD progressively develops, whilst in others, there is a loss of previously developed abilities, usually between 18 and 24 months [9]. The first study to examine the onset of ASD symptoms was based on an analysis of children diagnosed with autism who were video recorded at home before the diagnosis was made. The results of the study indicated that, at the age of 6 months, the symptoms of ASD cannot be noticed or are not very noticeable, and at the age of 6–12 months the symptoms of ASD are clearly visible in most babies: lack of response to their names, poor eye gaze, decreased shared attention, and narrow usage of deictic gestures. An increasing number of authors state that these sensorial and motor deviations and deviations in emotional modulation that occur between 18 and 24 months of age are early and often neglected symptoms of ASD [10], which precede socio-communication disorders and restrictive behaviors that more clearly indicate autism spectrum disorder [11].

Youngsters with severe ASD symptoms have more social deficits in communication and interactions, and they show increased restricted and repetitive behaviors compared to children with milder ASD symptoms [12]. Some of the core symptoms of ASD include atypical social and communication development [1]. Other ASD symptoms occur before the child’s second year of life and persist throughout life. These symptoms include a lack of emotional reciprocity, spontaneous seeking of joint interests, enjoyment, and affect. In addition, children with ASD have impaired facial emotion recognition, which is an early development of social skills in children without ASD [13]. Children with autism aged 3–5 show reduced and limited understanding of social context, lack of emotional reciprocity, non-verbal communication, and spontaneous behaviors (such as making contact with others or motor imitation of others) [14]. The theory of mind attempts to explain these clinical behavioral symptoms as a result of an important cognitive mechanism that results in the inability to understand and predict the feelings of other people, postulating a ‘primary’ impairment of cognition, specifically in the social domain [15]. Other symptoms that are observed at an early age are restricted and repetitive behaviors, which are one of the major concerns of the parents whose children are diagnosed with autism later on [16].

Restricted and repetitive behaviors are well-known symptoms of ASD but their development and trajectory are not fully clarified [17]. Repetitive behaviors can be detected at early ages, often before deficits in social communication [16]. Repetitive behaviors can be seen in typical development and have the purpose of mastering a developmental skill, and once the developmental skill is gained, these repetitive behaviors disappear. On the other hand, repetitive behaviors in ASD do not reduce over time and influence development [18].

Language deficits and delays in language development are typical for children with ASD and can vary significantly from child to child. In a study by Buzhardt, Wallisch [19], who followed the prelingual development of youngsters who were diagnosed with ASD later on, it was found that, at the age of 42 months, these children used fewer gestures compared to the control group, while vocalization was more frequent (without word production). These results indicate noticeable deviations in the speech–language development of youngsters with ASD at an early age. Highly functional individuals with ASD can have normal or high verbal IQ and structurally and grammatically adequate sentences. In contrast, low-functioning individuals can have agrammatic sentences, only use phrases, or never develop any language at all, but most children with ASD range between the high-, and low-functioning ends of the autism spectrum, meaning that many have semantic, syntactic, and phonological deficits [20]. In contrast, the pragmatic use of language is consistently compromised in all children with ASD [21]. Pragmatic can be defined as adequate use of language in the social context, and competence for pragmatic use in communication includes the capacity of the speaker to change the linguistic register according to a particular situation [22]. Given that children with ASD not only have language deficits but also a lack of social and communication deficits, it is difficult for these children to adjust their speech to the appropriate social situation, which makes their daily communication and functioning more difficult. Observing language from a developmental perspective, the continuity hypothesis [23] was developed, according to which pre-linguistic communication has a pragmatic function similar to that of early language and is, therefore, considered a significant precursor to further language development in children. This hypothesis suggests that children with stronger social motivation develop gestures and vocalizations, which will be the foundation for the development of early words. On the other hand, the speech attunement framework [24] suggests that children with strong social motivation seek language stimulation and, therefore, create the semantic basis for early language development. The foundations of these two hypotheses can be applied in the treatment of youngsters diagnosed with ASD by insisting on intentional communication as a link between social motivation and functional language. Puerto, Aguilar [25] points out that an accurate and early diagnosis of ASD is a precondition for including the child in an appropriate treatment program. Early intervention involves behavioral, cognitive, educational, and developmental approaches [26] for working with children and involving parents in the therapy process to stimulate the child’s development and generalize their abilities by relying on brain neuroplasticity. According to Kolb and Gibb [27], early treatment is based on the neuroplasticity of the brain conditioned by experience in which neural connections are generated and assembled, and the learning process happens as a result of the child’s interaction with the surroundings. For this reason, the parents’ engagement in the process of the early intervention model is to provide the child with the opportunity to learn through the performance of daily routines [28], develop communication skills [29], acquire knowledge, and generally learn through their own experiences. The early start of intensive treatments is of great importance for the progress in children with ASD [30]. Therefore, numerous authors suggest that the best time for treatment onset is between the ages of 1 and 3 [31,32]. In the last decade, there has been an increasing amount of literature regarding the outcomes of early intervention on the development of youngsters with ASD. According to Granpeesheh, Dixon [33], applied behavioral analysis (ABA) has a greater effect in youngsters who began treatment at a younger age compared to older children. Zhou, Xu [29] studied the effect of the intensive Early Start Denver Model intervention model on toddlers aged 18–30 months diagnosed with ASD. The results showed progress in language and social skills in children and a reduction in stress in parents.

Shi, Wu [34] showed that the majority of children with ASD, in whom early comprehensive treatment models (CTM) were used, achieved progress in reducing autistic symptoms and enhancing language skills. However, the same authors pointed out that the achieved results of these children still deviate from typical development, particularly concerning functional adaptive skills. Additionally, these authors emphasize that certain aspects of the treatment have a significant effect on its outcome, primarily: the approach to the child, the therapist conducting the treatment, the intensity of the treatment, and the total number of hours spent in the treatment.

### Present Study

Due to the increased ASD prevalence, the complex clinical picture, and the overlapping symptoms, we consider it crucial to focus on early interventions to reduce autistic symptoms and language deficits. Autistic symptoms and language deficits occur at an early age and affect other areas of development, such as cognitive, socio-emotional, speech and language, and motor skill development. The exploration of specific age effects on therapy results may contribute to further investigation and improvement of therapies for ASD.

The goal of the present study was to examine if early intervention is more effective in the reduction in autistic symptoms and language deficits in children 36–47 months old in comparison to children 48–60 months old.

The second goal was to examine if early intervention is equally effective for analyzed aspects of ASD (repetitive behaviour, social interaction, social communication, and emotional relations) within and between groups.

## 2. Methodology

### 2.1. Study Design

The mixed research design 2 (children’s age: 36–47 vs. 48–60 months) ×2 (time: T1 vs. T2) was adopted to examine differences in the effect of integrative therapy. The study was approved by the Institutional Review Boards (or Ethics Committee) of the Research & Development Institute “Life Activities Advancement Center” (EK-4/20 date: 28 October 2020).

### 2.2. Sample

The study sample consisted of 29 children from Serbia, Bosnia and Herzegovina, and Montenegro. All children were admitted to the Institute for Experimental Phonetics and Speech Pathology “Đorđe Kostić“ in Belgrade for integrative therapy during 2019. All the children had previously been diagnosed with ASD by a psychiatrist. The exclusion criteria were neurological disorders and vision, hearing, or motor impairment.

Firstly, we used the following inclusion criteria for entering the study: an established diagnosis of ASD and age of 36 to 60 months, after which children were continuously assessed for exclusion criteria until 20 participants were found for the G1 group (children 36 to 47 months old) and 20 participants were found for G2 group (children 48 to 60 months old). Evaluation for exclusion criteria was performed for 73 children, and 40 children started the intervention process (Figure 1). During the following year, 11 children dropped out of the therapy. The final number of participants included in the further analysis was 29 (Table 1). Groups were balanced for performance IQ, which was assessed using the Binet–Lezin scale [35].

### 2.3. Instruments

The Gilliam Autism Rating Scale, third edition (GARS-3) [36] was used to assess the presence of autistic symptoms. It is a questionnaire based on the DSM-IV [37] that can be administered by parents and caregivers. It contains six subscales: repetitive/restrictive behaviors, social interaction, social communication, emotional reactions, cognitive style, and non-functional speech. Given that the cognitive style and non-functional speech subscales should only be administrated to verbal individuals, for the purposes of our research, we used four subscales only. The whole questionnaire contains 58 items; due to the exclusion of two subscales, we used 44. For scoring, four-point Likert-type scale was used, ranging from 0 = never observed to 3 = frequently observed. The standard scores were obtained from the raw scores by conversion for each subscale. The sum of the standard scores shows the Autism Index (AI score), that is, the probability of the presence of autism, as well as the severity of the disorder, where autism is unlikely to be present when score is equal to or less than 54, and likely to be present-severity 1 when the score is between 55 and 70. A very likely presence of autism-severity 2 is indicated by a score from 71 to 100 and severity 3 by a score equal to or higher than 101.

For the assessment of speech and language development, we used the subscale Estimated Speech and Language Development (ESLD) of Scale for evaluation of psychophysiological abilities of children [38,39,40,41,42,43]. The ESLD includes the evaluation of receptive language development, expressive language development, syntax level, use of different types of words, morphological structure, receptive and expressive vocabulary, and pragmatic skills. The Scale for evaluating children’s psychophysiological abilities contained subscales that correspond to different years of age in relation to developmental norms for that age. The number of successfully performed items on the age-appropriate subscale was calculated relative to 100%. By applying a subscale specific to chronological age, we estimated the level of speech and language development and determined the relative speech and language development (RSLD) for each child separately. The formula: RSLD = (CA-ESLD)/CA was used for calculating RSLD. CA is the chronological age at the time of testing and ESLD is the estimated age based on the level of speech and language development obtained with the test. These measures are regularly used in speech and language clinical practice in Serbia [38,40,41,42,43,44,45].

The GARS-3 and ESLD were administered by a speech–language pathologist at two time points. The first time after the first week of therapy sessions, and the second time after one year from the start of therapy.

## 3. Intervention

The participants were admitted for a multidisciplinary assessment at the Institute for Experimental Phonetics and Speech Pathology “Đorđe Kostić“ (IEPSP). The KSAFA system [42,46,47,48] was developed and was used for assessment and therapy conduction at the IEPSP, where a multidisciplinary admission team (speech–language pathologist, psychologist, psychiatrist, and pediatrician) performed an examination. The process started with the inspection of medical records and an interview with parents, followed by child observation and the application of diagnostic tests. The examination was first performed by the whole multidisciplinary team and then by the individual assessment of each team member. After the assessment process, the multidisciplinary admission team decided on a multidisciplinary implementation team that created an initial integrative therapy treatment plan. The multidisciplinary implementation team for all participants included: a speech–language pathologist, special educator–sensory integration therapist, special educator–occupational therapist, and psychologist.

With the aim of optimal development and synchronization of speech, motor, sensory and social functions, the therapeutic procedures carried out within the integrative therapy were individualized, based on written work instructions, and could be interpreted orally to the team members that participated in the therapy or to the parent, as well as being repeated and/or redefined depending on the needs. The treatments were daily and were synchronized on a daily basis, respecting the individual dynamics of the development of functions and correlating them with each other. Each child had 16 h of treatment weekly, with the adaptation of spatial and didactic conditions to the child’s needs/preferences in visual and auditory terms. The speech–language pathologist was the coordinator of the team, responsible for the course and monitoring of the therapy. The therapy process contained the following segments:

Speech–language therapy: conducted by the speech–language pathologist. The goal was a development of verbal and non-verbal forms of communication. Therapy focus was aimed at encouraging interaction for communication, development of auditory attention, enrichment of receptive and expressive vocabulary, concept development, sentence development, development of adequate emotional speech expression, development of morphology (dynamics for the Serbian language), encouraging spontaneous speaking, and spoken pragmatics with the aim of verbal interaction and communication. Speech–language therapy was conducted twice a day, for 60 min, individually, in a 20 m^2^ room equipped with a table, chairs, toys, and other didactic materials. During the last 5–10 min of the therapy session, the parent received a work report, instructions, and tasks for the day.

Sensory integration therapy: conducted by the special educator–sensory integration therapist. The goal of the therapy was to gain independence in the integration of the sensations that the child receives from their body or environment in order to independently participate in daily activities and social environment. Therapy implies that the child is exposed to various auditory, visual, tactile, vestibular, proprioceptive, and olfactory stimuli through play. The therapy was conducted in a room equipped with: a swing, a crawling tunnel, mats, pillows, balls of different sizes, tactile balls, massage levers, equipment for climbing and descending (climbing ladder, slide), vibrating, and musical toys. Sensory integration therapy took place once a week, for 60 min, individually, with the child and the parent, usually the mother, led by a special educator.

Occupational therapy: conducted by the special educator–occupational therapist. The goal of therapy was to help the child to elevate and develop skills that will improve their participation in social and everyday activities. It was a group treatment focused mainly on dressing, feeding, and personal hygiene through play and imitation, as well as on other everyday activities. Occupational therapy took place in a large room, equipped with a toilet, a blackboard, a climbing ladder, benches, various sets of toys for stimulating different social activities (kitchen, farm, garden, workshop, and market), and several small tables and chairs. It took place in a group of 5–7 children. The therapy was conducted by two special educators for an hour, five times a week.

Family counseling: conducted by the psychologist. The aims of these meetings were: accepting the intervention program, providing support for the child in the home environment, regulating the child’s basic habits (eating, sleeping, potty training, etc.), reducing parental stress, setting healthy boundaries, accepting the child’s condition, and preventing secondary effects of primary developmental problem. The psychologist and licensed family counselor conducted advisory sessions with parents two times per month for 60 min. Each session was conducted without the presence of the child, with an individual parent or couple, depending on the present family situation. Advisory sessions were administered through using family systems therapy interventions.

The team coordinator was a speech–language pathologist. The assessment team analyzed the progress every three months, discussed the progress with team members and parents, and, if necessary, provided guidelines for individual segments, further course of treatment, and results, as well as suggested improvements to segments of the integrative therapy treatment plan, with the possibility of changing any of the members of the implementation team.

### 3.1. Procedure

After the multidisciplinary admission team assessment, the children were included in the study based on the inclusion and exclusion criteria. Treatment was started no later than seven days after the assessment. After 10 days of therapy, the first assessment with the GARS-3 and ESLD was performed. The assessment was repeated after 12 months of treatment.

### 3.2. Statistical Analysis

Descriptive statistical measures were used to describe and analyze the samples. In order to calculate the differences between the two groups of children in terms of performance IQ, a *t*-test for the independent samples was used. To calculate the differences in the Autism Index 1 and Autism Index 2, a paired sample *t*-test was used. The differences between the two groups in reducing autistic symptoms and speech–language development at two time points were calculated by using repeated measures ANOVA.

## 4. Results

Our results of GARS-3 regarding the Autism Index for the total questionnaire and each subscale are given in Figure 2a and Figure 3 (see also Appendix A). At the first testing, there was no statistically significant difference between the groups (*p* > 0.05) for the total questionnaire and all subscales. In the second test, there was a statistically significant difference for the total questionnaire and all subscales separately (*p* < 0.05). There was a statistically significant difference between T1 and T2 for the total questionnaire, repetitive behaviour, social interaction, social communication, and emotional reactions for G1 and G2 (*p* < 0.05).

Our results regarding the effect of the group on the difference in Autism Index at T1 and T2 showed that there was a statistically significant effect of the group on the total questionnaire, repetitive behaviour, social interaction, and social communication subscales (*p* < 0.05; Figure 3). Results for emotional reactions do not show a statistically significant difference (*p* = 0.068; Figure 3).

Figure 2b presents the results of the ESLD in the first and second tests. Our results showed no statistically significant difference between the groups at the first testing (*p* < 0.05). In the second test, there was a statistically significant difference between the groups (*p* < 0.05). When we analyzed the effect of the group on the ESLD results at T1 and T2, our results showed that there was no statistically significant difference (*p* = 0.626, Appendix B).

## 5. Discussion

In this study, we examined whether there is a difference in the effect of early intervention on autistic symptoms and speech–language deficits between children with ASD who started therapy at 3 years of age and children at 4 years of age. We examined whether there was a difference between younger children who started integrative therapy at the age of 36–47 months and the older group of children who started integrative therapy at the age of 48–60 months. Our results showed that early intervention had a better effect on autistic symptoms in younger children, but there was no difference among the groups concerning language abilities. Our second goal was to examine if early intervention is effective for all observed aspects when we observe each group separately, and our results showed that children 48–60 months old did not show improvement in social communication.

The results of our study suggest that autistic symptoms are reduced more in younger children than in older children. Previous studies have shown that early interventions have a positive effect on development in youngsters with ASD [49,50]. Dawson, Rogers [49] highlighted the long-term effect of early intervention in children who were included in integrative therapy and ended it 2 years prior; there was a lower degree of delay in terms of adaptive behavior compared to the control group. Although the above studies examined the effects of early intervention, they did not investigate the impact of age within the period of early development. We know that early intervention has positive effects [50,51], and our results showed that even with an age difference of one year, that is, between 3-year-old and 4-year-old children, there is a difference in the treatment effect. These results could be explained by the existence of sensitive periods in brain development in which certain experiences (stimuli) have lasting effects on the brain and behavior [52]. During this period, the brain is more sensitive to sensory inputs to which it is exposed through interaction with the environment, which is why this period is considered optimal for learning. During this period, neural representations are narrower than those widely adapted for relevant stimuli to be more precise and begin to respond to stimuli to which they are frequently exposed, thus enabling more accurate and efficient processing of dominant and frequency stimuli (information) [53]. Therefore, early intervention is based on neuroplasticity conditioned by experience as one of the basic features of the brain (nervous system), which implies that learning is the result of a child’s interaction (experience) with the environment [27].

When we observed our results separately for individual elements of autistic symptoms, such as repetitive behaviors, social interaction and communication, and emotional reactions, our results also showed that early intervention had better effects in younger children.

Repetitive and stereotypical behaviors are considered the earliest ASD symptoms observed even at the age of 12 months [54]. Leekam, Prior [55] concluded that this kind of behavior is connected to attention, learning, academic achievement, and social interaction. Our results showed that early intervention reduced repetitive behavior in both groups but that the effect was stronger in younger children. Chen, Yoder [56] found a reduction in repetitive behaviors in youngsters with ASD who lack communicative speech after the application of specific integrative therapy. They also found a relationship between repetitive motor behaviors and other areas of development. Repetitive behavior and somatic and/or sensorial manipulation of objects is also related to poor adaptive skills and lower non-verbal and verbal IQ [57]. Given that repetitive and stereotypical behaviors can be expressed at an early age [58] and have an impact on various aspects of development [59], the importance of reducing these symptoms is emphasized, and our results point out that even a one-year-long intervention can have a significant effect on these symptoms.

In our research, the greater progress in younger children was also noticed in terms of social interaction and social communication. It is known that social interaction and communication are based on the integrative and simultaneous processing of information obtained by analyzing the non-verbal behavior of the interlocutor (social partner) during joint attention to attract the interlocutor’s attention or respond to the interlocutor’s solicitation of attention [60]. Through shared attention, the child gains experiences with the objects and events that surround them, discriminating between them and giving them a social reference with the help of parents who recognize the moment when the child’s attention is engaged to a sufficient level to start the learning process [61]. It is for this reason that shared attention is the basis for cognitive, social, and speech–language development [62]. However, children with ASD often have altered sensory and motor functioning and their interaction with the environment; therefore, the acquired experiences are qualitatively and quantitatively altered. This means that in the social interaction between partner and child or social interaction between partner, object, and child, these children have a reduced, incomplete opportunity to acquire social and communication experiences [63]. Mutual gaze in infant–parent interactions may activate the social brain network [64]. Children with ASD, around two and six months of age, do not make eye contact with caregivers, which deprives them of their social experience. A meta-analysis research that studied the outcome of early intervention on the social communication abilities of toddlers diagnosed with ASD showed that children in early interventions had significantly higher improvements in social abilities than the control group [65]. Many studies have studied the association between shared attention and later speech–language development. One such study is that of Kasari, Paparella [66] who showed that children who had early intervention focused on improving skills of social communication had longstanding effects on language skills compared to children in the control group. Additionally, a link between joint attention skills and language outcomes had been found in correlational studies that have found significant associations between joint attention skills and language outcomes [67]. We find it important to highlight that in our study, the older group did not express improvement in social communication, while an improvement was observed in the younger group. Furthermore, when it comes to speech–language abilities, which are, as we discussed before, closely connected to social communication skills, both groups were at the same level at the start of the therapy process but also had the same level at the end of the experiment; thus, the results cannot be explained by the influence of the speech–language development level. These results may indicate the importance of therapy modifications and paying special attention to this developmental aspect in older children. Socio-emotional development implies the ability to express and process positive and negative emotions, achieve interpersonal relationships, and explore the environment in order to learn [68]. Many studies have indicated that early socio-emotional atypicality can predict later disorders in social and emotional control, such as poor levels of regulation, high levels of negative affect, and distress [69,70]. These impairments represent the core of ASD symptoms. The results of our study indicated a stronger reduction in these symptoms in children who started integrative therapy at an earlier age. These results are significant because according to some theories, an adequate level of emotional development can compensate for atypical (or inadequate/subnormal) intellectual functioning and adaptive behavior and provide insight into a person’s inner experiences and self-regulating abilities [71]. In addition, information about the emotional profile of the child can be of great importance for understanding his/her behavior and creating individualized treatment and support [72].

Atypical language development, in terms of delayed expressive language, is one of the first symptoms of ASD in children. Infants who go on to be diagnosed with ASD in the future show a lack of bubbling [69] and vocalization in interaction with adults [73]. It is known that the production of words with meaning by 24 months [74] and the number of words produced by 2.5 years [75] are important predictors of language, cognitive, and adaptive behavior abilities for youngsters with ASD. Therefore, many studies emphasize the importance of early interventions for children with ASD, especially interventions that target language skills to affect children’s developmental potential. There are two types of therapy approaches for youngsters diagnosed with ASD that focus on spoken language: targeted and comprehensive interventions. Targeted interventions focus on developing prelinguistic and communication skills. These interventions focus on developing expressive and receptive language and focus on its implementation in social aspects, together with gestures and other non-verbal communication. Comprehensive interventions have many goals and outcomes because they target a wide range of skills and behaviors, such as cognitive, motor, and self-care skills; play; and language production. Early interventions should combine these two approaches so that the effects of the treatment are as effective as possible and have the longest possible effect. The results of our research indicate the necessity of recognizing the earliest deficits in communication in infants and the importance of early inclusion in the therapy process to maximize the child’s potential. Although our results showed that younger children had better speech and language development after one year of integrative therapy, no statistically significant difference was found. Considering that speech and language development in our sample was profoundly impaired, a period of one year was perhaps too short to expect significant progress in this segment of development. Another possible explanation for our results is that we have observed all aspects of language development together, but there is a possibility that differences can only be seen in certain aspects.

The limitation of our study was that we measured language development in general, but an assessment of different aspects of speech–language development separately could be more informative. Future studies should consider a sample of a larger group of youngsters, as well as a control group of youngsters without the treatment to exclude the natural development effect. It would also be recommended to conduct longer longitudinal follow-ups of children to determine whether the effects of treatment are visible for a longer period, especially when it comes to language development.

## 6. Conclusions

Youngsters diagnosed with ASD exhibit altered behavior in the form of repetitive patterns, lack of social interaction, and socio-emotional reciprocity [1]. Speech–language development in these children is also deviant and delayed [19]. Many studies have addressed the significance of early intervention for autism symptoms [76,77]. Our study emphasizes the importance of age, even within the period of early development, on the effect of therapy on ASD symptoms, such as repetitive behavior, social interaction, social communication, and socio-emotional development. The difference of one year of age at the time of starting treatment between children aged three and four years is not crucial for speech–language development viewed as a whole, but future research that would follow specific aspects of speech–language development could indicate some differences. Our study highlights the importance of emphasizing the exact age when using the terms “early intervention” and “early development” in future studies and practice because it is necessary to determine and make guidelines about which particular ages are crucial for starting treatment in certain developmental aspects. Social communication represents a special challenge for children four years and older, and thus, in the process of therapy methods creation, special attention should be paid to this developmental aspect.

## Figures and Tables

**Figure 1 children-10-00122-f001:**
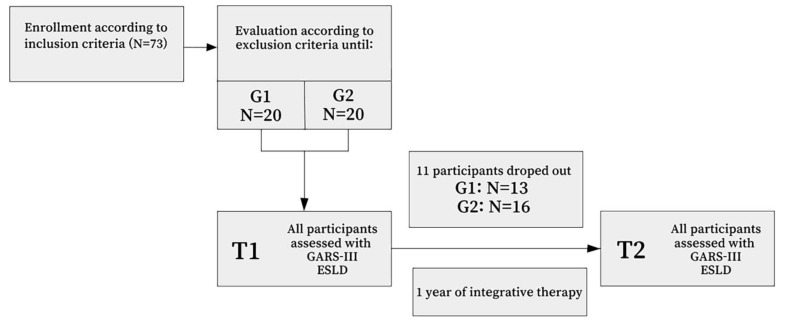
Flowchart with the selection of participants.

**Figure 2 children-10-00122-f002:**
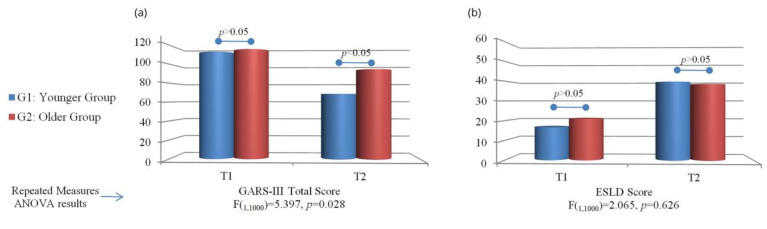
Average scores on (**a**) GARS-III and (**b**) ESLD for G1 and G2 at two time points and repeated measures ANOVA results.

**Figure 3 children-10-00122-f003:**
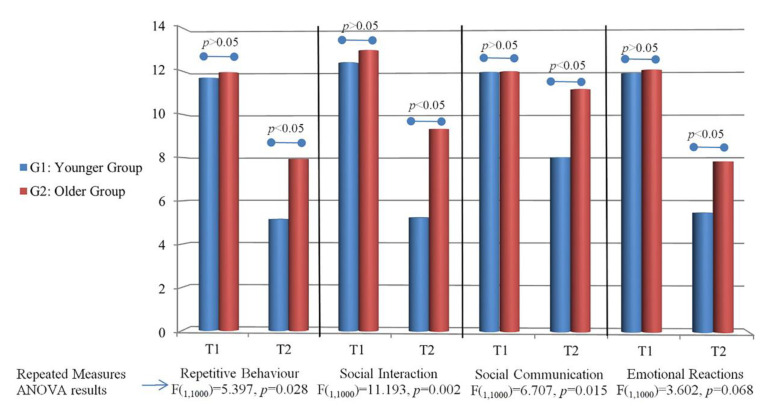
Average scores on each GARS III subscale for G1 and G2 at two time points, and repeated measures ANOVA results.

**Table 1 children-10-00122-t001:** Participants’ characteristics.

	No.	Gender	Age Mean ± SD	IQ Mean ± SD
G1	13	M = 9; F = 4	41.08 ± 3.57	73.10 ± 9.51
G2	16	M = 11; F = 5	54.37 ± 3.37	70.50 ± 16.01
				*p* = 0.652

## Data Availability

The data that support the findings of this study are available from the corresponding author, NS, upon reasonable request.

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
