# Peer review of "Importance of Early Intervention in Reducing Autistic Symptoms and Speech–Language Deficits in Children with Autism Spectrum Disorder"

_children, 2023, doi:10.3390/children10010122_

Round 1
Reviewer 1 Report
The paper is promising for its topic and in line with the aims of the Journal.
In particular, the study focuses on the precocity of the intervention for children with ASD. The explored issue is: At what age to start early intervention? When is early intervention most effective in modifying the developmental trajectory/ASD symptoms?
Despite these interesting premises, the work overall is very weak.
The authors choose a mixed research design 2 (children’s age: 36-47 vs 48-60 months) x 2 (time: T1 vs T2 12 months after treatment), but there is no a specific section in the manuscript to describe it. Similarly, a specific section presenting study aims and testing hypothesis is missing in the paper.
The sample of children with ASD is small, but this limitation could be justified by the fact that it is a clinical sample.
In the opinion of the referee, however, the methodological structure of the research is globally very weak, but its limits cannot be improved since they derive from previous methodological authors’ choices.
The methodological weak points are the following:
a) The study focus is on verifying the effectiveness of early intervention, but four treatments are implemented simultaneously. Therefore: to which intervention to attribute the improvement?
2) The treatments are not operationalized (what objectives? Have they been manualized, so are they replicable?).
For example, in the introduction authors discuss different approaches for language treatment (see lines 117 and subsequent) and parental involvement (lines 127 and subsequent) but they do not specify in the method the chosen approach interventions. Authors simply limit to quote the professionals who conduct the different programs (for example: speech-language pathologist, special educator etc ). This multidisciplinary approach makes it even more difficult to answer the question (implicit to research hypothesis): which early intervention is most effective with younger ASD children? The question is very important for its practice and prevention (individualization of early intervention/indications of the effectiveness and costs of the programs).
3) Improvement of speech-language deficits is a crucial variable chosen by the authors to evaluate the outcome of the intervention(s). However, the authors use a global measure (ESLD), not a tool aimed to capture specific delays/deficits in the different areas of language and communication. Although the authors recall this limitation in the conclusions (line 387), this methodological choice greatly weakens the quality of the study and it cannot be improved.
Other minor weak points are:
a) Abstract. Too long, it can be reduced.
b) Results. Differences tested by t-test did not result statistically significant, but authors comment that a trend can be observed. This conclusion is incorrect and is not supported by statistical inferential test.
c) References. The list is too long. Authors should choose less generic articles (that is, references more closely related to the study topic) and select only the most recent ones.
Author Response
Dear Reviewer,
Thank you very much for your comments and suggestions. Your comments really helped us to improve our paper. Please see the attachment for the file with our response.
Kind regards,
Nina Stanojević

Reviewer 2 Report
Importance of Early Intervention in Reducement of Autistic Symptoms and Speech-Language Deficits in Children with Au-tism Spectrum Disorder.
It is an important topic, but some considerations must be performed related to methods:
- More definitions of the sample should be included.
- A flowchart with the selection of participants should be included.
- Definition of the intervention (in this form is not enough) it is not possible to understand.
- Results should be appear in a clearer way
Author Response

(The authors gave the same response as above.)

Round 2
Reviewer 2 Report
Thank you very much. All comments have been answered.